# A Stochastic Model and Investigation into the Probability Distribution of the Thickness of Boride Layers Formed on Low-Carbon Steel

**Julio C. Velázquez-Altamirano [1,\*], Itzel P. Torres-Avila [2], Gerardo Teran-Méndez [1], Selene I. Capula-Colindres [3], Roman Cabrera-Sierra [1], Rafael Carrera-Espinoza [4] and Enrique Hernández-Sánchez [2,\*]**

[1] Departamento de Ingeniería Química Industrial, ESIQIE, Instituto Politécnico Nacional, UPALM Edif. 7, Zacatenco, 07738 México City, México; gerardoteranm@gmail.com (G.T.-M.); roma_ipn@yahoo.com (R.C.-S.)

[2] Departamento de Bioingeniería UPIBI, Instituto Politécnico Nacional, Avenida Acueducto s/n Barrio La Laguna Ticomán, 07340 México City, México; itzelpam_9318@hotmail.com

[3] Laboratorio de Microtecnologia y Sistemas Embebidos, CIC-IPN, Av. Juan de Dios Bátiz s/n, Col. Industrial Vallejo, 07738 México City, México; selenecapula@gmail.com

[4] Department of industrial and mechanical engineering, Universidad de las Américas Puebla, Ex Hacienda Sta. Catarina Mártir S/N. San Andrés Cholula, 72810 Puebla, México; rafael.carrera@udlap.mx

\* Correspondence: jcva8008@yahoo.com.mx (J.C.V.-A.); enriquehs266@yahoo.com.mx (E.H.-S.);
Tel.: 52-1-55-3939-0712 (J.C.V.-A.); 52-1-55-1069-2992 (E.H.-S.)

**Abstract:** The stochastic nature of the thickness of boride layers formed on carbon steel is described in this paper. Additionally, the probability distribution of the layer thickness is studied to determine the best-fit probability distribution. The study combines the use of an empirical model (power-law) and the Markov chain principles, with the purpose of demonstrating that it is feasible to develop a model that represents the non-uniformity of the thickness of boride layers that form on carbon steel. The results indicate that the mean and variance tend to increase when the time or temperature is increased. The findings of this paper demonstrate that an analytical solution to the Kolmogorov's system differential equation can adequately represent the behavior of non-uniform boride layer formed on low-carbon steel, regardless of the temperature or duration of treatment.

**Keywords:** steel; modeling studies; boride layers; probability distribution

---

## 1. Introduction

Boriding is a thermochemical treatment that is used to obtain hard layers. With this process, it is possible to enhance the wear and erosion resistance of metallic materials [1]. The small size of the boron atom plays a crucial role in the ability of boron to diffuse into metals or alloys such as iron, carbon steel, stainless steel, nickel, and titanium [2]. In carbon steels, iron-boride formation entails a process of nucleation, in which the particles are formed on the metallic surface to induce layer formation through the process of diffusion. It is necessary to study and model this process of layer formation, especially when automation is implemented and optimization is required. In this context, this paper describes the non-uniformity of the thickness of boride layers, as obtained by implementing the boriding process on carbon steel (AISI 1018). Considering this, the layer thickness can be modeled as a stochastic process. To simplify this introduction, details about the boride layer formation and mathematical foundations have been separated into sub-sections.

### 1.1. Boride Layer Formation on Carbon Steel

To improve the wear and erosion resistance of metallic materials, it is necessary to apply surface hardening treatments such as the boriding process. Boriding is a thermochemical process in which boron atoms diffuse into a metal surface because of their small size and high mobility at high temperatures; the result is an extremely hard boride layer. [2]. In addition, applying the boriding method to a metal can extend its service lifetime by up to 10 times that of the same metal that was not subjected to boriding [3,4].

Boriding is most suitable for carbon and low-alloy steels, and is typically performed under the conditions of temperatures ranging from 800 to 1050 °C, and treatment durations ranging between 1 and 12 h [4,5]. When boriding is applied to carbon steel, the generated layer has a sawtooth-like shape, and may contain one iron boride phase ($Fe_2B$), or have a dual phase ($FeB + Fe_2B$). Typically, FeB is harder than $Fe_2B$, but an iron boride phase is not desirable because of its higher brittleness.

To form a suitable $Fe_2B$ layer, a continuous flux of boron is needed. Assuming that the growth of the $Fe_2B$ boride originates at the external surface and proceeds toward the interior of the substrate, the boron flux has to move across the previously formed $FeB/Fe_2B$ diffusion barrier. Figure 1 illustrates how the dual-phase layer is formed. First, after the substrate comes into contact with the boron source, $Fe_2B$ nucleation begins. Typically, the proportions of Fe and B do not create conditions that are favorable for sustained $Fe_2B$ layer formation. The proportion of boron tends to be lower because the active boron accumulates at the outermost zone of the $Fe_2B$ layer. This sequence of actions explains why the FeB is typically found on the outer layer in carbon steels that are subjected to boriding for relatively long periods, and at high temperatures. Therefore, the thickness and quality of the boride layer are dependent on the chemical composition of the substrate, boron potential of the boron source, temperature, and treatment duration [1,6]. Thus, the morphology, growth, and phase composition of the boride layer are affected by the alloying elements contained in the substrate [7], which can hinder the boride diffusion process.

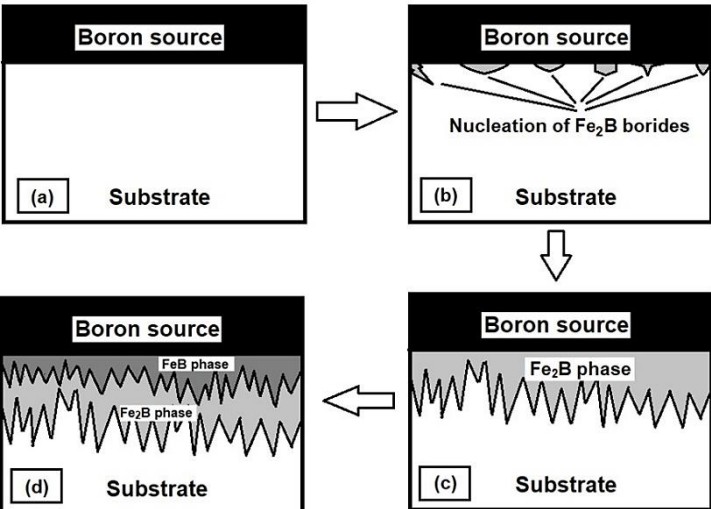

**Figure 1.** Schematic representation of boride layer formation where (**a**) represents the initial state of the process, (**b**) represents the second stage where the nucleation of the iron borides begins, (**c**) represents the third stage where the Fe2B phase is consolidated and (**d**) represents the last stage of the process, where the FeB phase is formed on the surface of the material.

To control, automate, and optimize the boriding process for a specific industrial installation, knowledge of layer growth modeling is essential [2,8]. According to several published papers [1,2,8–10], it is reasonable to suggest that boride layer growth has a stochastic nature, meaning that it occurs randomly. This phenomenon can be observed when the thickness of the boride layer on carbon steel is measured. The layer does not have a uniform thickness, as illustrated in Figure 1 in this

paper. Moreover, the sawtooth-like morphology of the boride layers can be observed in Figure 1 of reference [11], Figure 2 of reference [12], and Figure 8 of reference [1]. All of that Figures illustrate a classical sawtooth-like morphology, with its non-uniform layer thickness that is dependent on the treatment duration. This indicates that the frequently used classic deterministic models could be somewhat limited. For this reason, the probabilistic nature of this phenomenon was studied by using a Markov chain model.

### 1.2. Mathematical Foundation

For researchers in the field of materials engineering, it is common to study the stochastic nature of the chemical, electrochemical, mechanical, and thermodynamic behaviors of materials [13–17] because of their inherent complexity. Many properties and characteristics of metals, such as the alloy composition, microstructure, surface roughness, temperature, treatment duration, and chemical composition of the surrounding media, all influence these aforementioned behaviors. Such complexity mandates the development of advanced mathematical models and simulation tools to obtain a better understanding of the target material behavior, and more accurate predictions and estimations of how material properties change in a specific environment.

Several studies have been dedicated to develop a deterministic approach to investigate the fundamental physico-chemical mechanisms of boride layers formed on various steels. Campos-Silva et al. [18–20] used kinetics and diffusion models to obtain and present a relatively simple explanation of boride layer growth; their kinetics and diffusion models are deterministic. Hernández-Sanchez and Velázquez also proposed the use of an empirical model based on a power-law equation [1]. Even though it is of significant importance to understand the basics of microscale boride layer growth, deterministic approaches are rather limited by the complexity and unpredictability of the process. For example, the sawtooth-like shape of the boride layer demonstrates the stochastic nature of the process. However, Hernández-Sanchez and Velázquez mentioned that their method to estimate the layer thickness only yields the average boride layer thickness [1]. Considering this, a methodology that is frequently used to determine the average layer thickness is illustrated in Figure 2.

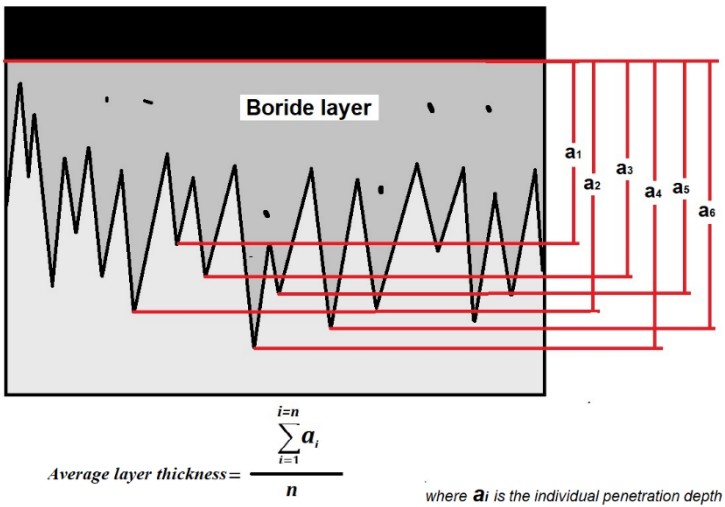

**Figure 2.** Methodology for layer thickness determination.

The kinetics of boride layer growth were studied using a well-known method based on the Arrhenius equation [1,2]. All diffusion processes involved in boride layer growth obey a power law, which is a solution of the Arrhenius equation [1], which is given as follows:

$$x^2 = Kt \tag{1}$$

where $x$ is the layer thickness, μm; $K$ is the growth rate constant, $μm^2·s^{-1}$; $t$ is the diffusion time, s.

It can be ascertained that the thickness of the boride layer is directly influenced by the square root of the diffusion time.

The growth rate constant $K$ is dependent on the treatment temperature, and this behavior can be described by the following Arrhenius mathematical expression:

$$K = K_0 \, \mathrm{Exp}\left[-\frac{Q}{\mathrm{R}T}\right] \qquad (2)$$

where $T$ is temperature, $K$; $K_0$ is a pre-exponential factor that is dependent on the boron potential of the boron source surrounding the steel substrate during the treatment, $m^2·s^{-1}$; R is the universal constant of ideal gases, $8.3144 \, J·mol^{-1}·K^{-1}$; $Q$ is the activation energy, $J·mol^{-1}$.

It is also possible to present Equation (2) in logarithmic form, as follows:

$$\ln K = \ln K_0 - \left[\frac{Q}{\mathrm{R}}\right]\frac{1}{T} \qquad (3)$$

Equation (3) describes the linear relationship between the natural logarithm of the growth rate constant and the treatment temperature. Equation (3) can be represented as a linear function, and it is possible to obtain the activation energy from the slope of the straight line of $\left(\frac{Q}{\mathrm{R}}\right)$.

Equations (1)–(3) are deterministic, and only serve to simplify the study of boride layer growth. As can be observed in Figures 1 and 2, the thickness of the boride layer on carbon steel is not uniform across the entire surface, even when the sample has been treated at the same temperature for the same duration. A heterogeneous surface microstructure and boron source could be the main causes of the sawtooth-like layer morphology. This means that the formation of the boride layer on carbon steel is inherently stochastic, with the random variable being the boride layer thickness. Intuitively, a random variable can be understood as a quantity with a value that is not fixed. Therefore, it can assume different values. A probability distribution is generally used to explain the probability of different values. In formal terms, a random variable is a function that is defined in a probability space [21]. Given a random variable, it is not possible to be certain of the value that will be obtained via measurement or calculation; however, it is known that there is a probability distribution that is associated with the set of possible values. Thus, when working with random variables, it is necessary to consider a large number of randomized experiments to optimize statistical analysis, and to quantify the results such that a real number is assigned to each of the possible resulting values obtained via the experiment. In this way, a functional relationship is established between the elements of the sample space associated with the experiment and real numbers [21].

In this study, the boride layer thickness can be considered a continuous random variable because its path is a non-countable set, meaning that the set of possible values for the variable includes the entire feasible range of real numbers [21,22].

When planning to analyze boride layer thickness data, initially it is necessary to investigate the distribution pattern-specific characteristics. This type of data can be visualized as a histogram, as shown in Figure 3. This histogram can be fitted to a theoretical probability density function, such as log-normal distribution, gamma distribution, or generalized extreme value (GEV) distribution.

The frequency distribution of the layer thickness parameter is presented in a discrete mode. This type of discrete function cannot by simply described by an equation. Therefore, to facilitate statistical analysis, it is common practice to adapt the discrete distribution pattern to the best-fit continuous distribution profile [23].

In materials science and engineering, the most commonly implemented probability distributions are the normal distribution [24], Weibull distribution [25], Poison distribution [26], gamma distribution [27], log-normal distribution [28], exponential distribution [17], Gumbel distribution [29,30], and GEV distribution [31].

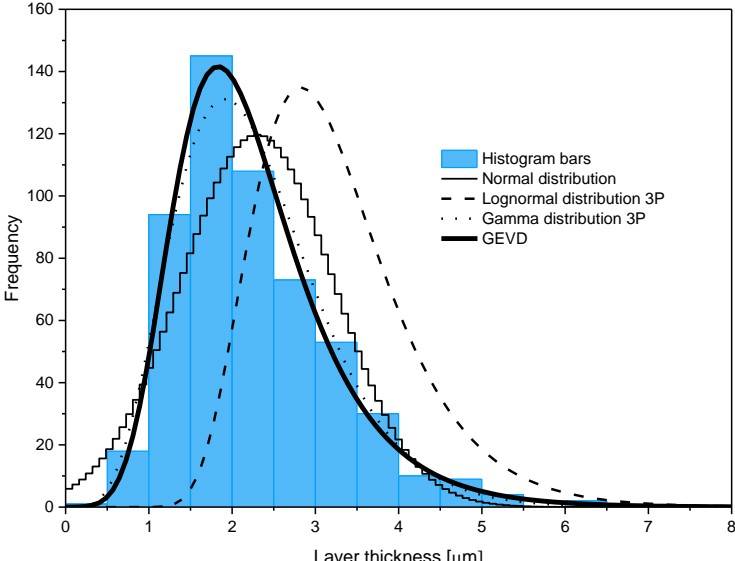

**Figure 3.** Histogram of observed boride layer thickness formed on carbon steel under the conditions of a 900 °C, 20 min treatment; the normal, log-normal, gamma, and GEV distributions are shown.

Since the boride layer forms a sawtooth-like shape on carbon steel, it is possible to measure each "peak" as a local maximum of boride layer thickness. Estimating the maximum boride layer thickness can help to determine with certain confidence, when the layer formed on carbon steel has a considerable thickness. GEV distribution is a well-known tool used to study the maximum boride layer thickness formed on carbon steel [32], and can be expressed as a cumulative distribution function as follows:

$$G(x) = \begin{cases} \exp\left(-(1+\xi z)^{\frac{-1}{\xi}}\right), \xi \neq 0 \\ \exp(-\exp(-z)), \xi = 0 \end{cases} \tag{4}$$

The following equations describe the probability density function (PDF):

$$g(x) = \begin{cases} \frac{1}{\beta}\exp\left(-(1+\xi z)^{\frac{-1}{\xi}}\right)(1+\xi z)^{-1-\frac{1}{\xi}}, \xi \neq 0 \\ \frac{1}{\beta}\exp(-z - \exp(-z)), \xi = 0 \end{cases} \tag{5}$$

where $z \equiv \frac{x-\alpha}{\beta}$; $\alpha$ and $\beta$ are the location and scale parameters, respectively, and $\xi$ is the shape parameter.

Each histogram obtained in this study was fitted to a GEV distribution and subsequently analyzed.

It is acceptable and common to use Markov chains to model a stochastic process, such as that in the case of a boride layer generated on low-carbon steel [33,34].

Caleyo and colleagues [33] proposed a model that implements a continuous-time, discretized, pure-birth, homogeneous Markov process to describe the external corrosion defects in buried pipelines. The term $p_i(t)$ was defined as the probability that the corrosion defect depth is in the ith state at any given moment, and time $t$ was determined according to the measured corrosion defect depth distribution at time $t$. The intensity $\lambda_i$ was related to an empirical equation that represents the corrosion defect growth as a time function. $p_{ij}(t)$ represents the probability that the process is in the state $i$ and "jump" to the state $j$ at some time; all this process obeys the following system of Kolmogorov's forward differential equations:

$$\frac{dp_{ij}(t)}{dt} = \begin{cases} \lambda_{j-1}(t)p_{ij-1}(t) - \lambda_j(t)p_{ij}(t), j \geq i+1 \\ -\lambda_i(t)p_{ii}(t), \end{cases} \tag{6}$$

The same mathematical process proposed by Caleyo et al. for the degradation of oil and gas pipelines was implemented in this study [34]; however, it was adapted for application to the boride layer thickness formed on carbon steel. In our case, instead of discretizing the corrosion defect depth, the boride layer thickness generated on the carbon steel was discretized and divided into $n$ number of states. Details on how to obtain the solution of Equation (6) for boride layer thickness are presented in subsequent sections of this paper.

## 2. Materials and Methods

### 2.1. Boriding Process

Commercial low-carbon steel AISI 1018, the chemical composition of which is provided in Table 1, was used as the substrate in this study.

**Table 1.** Chemical composition of AISI 1018 steel (wt.%).

| C | Mn | Si | P Max | S Max | Fe |
|---|---|---|---|---|---|
| 0.15 0.20 | 0.60/0.90 | 0.15/0.30 | 0.04 | 0.05 | Balance |

Bar-shaped AISI 1018 steel was sectioned into cylindrical samples with a diameter of 10 mm and a length of 4 mm. Before initiating the boriding process, the samples were ground with 100–1200 grit abrasive paper. Then, the samples were cleaned by performing sonication in ethanol for 5 min, rinsed with distilled water, and dried in hot air. The samples were embedded into a cylindrical case (AISI 304) containing the boron powder source (Hef-Durferrit). The samples were covered with 15 mm of powder on each side to prevent oxidation [35].

The boriding process was carried out at temperatures of 900, 950, and 1000 °C for a duration of 0.33, 1.5, 3, or 5 h in a conventional furnace that did not provide an inert atmosphere. After boriding, the samples were allowed to cool outside the furnace at room temperature.

### 2.2. Characterization of Borided Samples

After boriding, the samples were cleaned in an ultrasonic bath with ethanol and deionized water (50/50) to remove any contaminants remaining from the boriding process. The borided samples were cross-sectioned and metallographically prepared by gradually sanding each of them with SiC paper, and then polishing them with a 0.05 μm alumina suspension to achieve a mirror finish. Finally, the samples were etched by using a 3% Nital solution. The morphology of each boride layer was examined by using a Olympus GX51 optical microscope (Olympus, Center Valley, PA, USA) and scanning electron microscope (SEM) (JSM-6360LV, JEOL, JEOL LTD., Akishima, Japan) to obtain morphology, films thicknesses and elemental composition through energy-dispersive spectroscopy (EDS, JEOL LTD., Akishima, Japan). At least fifty measurements were performed from a fixed reference on different sections of the borided samples by following the methodology described in Figure 2. The boride layer thicknesses of each sample were measured by using Image-Pro Plus V6 image analysis software, and the mean thickness values are depicted in Table 2. The phases present in the boride layers were determined by the X ray diffraction technique (XRD), by operating a D8 FOCUS diffractometer (Bruker, Billerica, MA, USA) with Cu-K$\alpha$ radiation at a wavelength of 1.5418 Å.

The hardness of the boride layers was evaluated by performing Vickers microindentation with the aid of a CMS-CHV1M Vickers microdurometer (CMS Metrology, Queretaro, México). A total of 10 microindentations, each with a 100 g load, were perpendicularly applied to the boride surface layer according to the recommendations of the ASTM E-384 standard [36]. The microindentations were applied to the compact zone of each boride layer in consideration of the layer thickness as a function of the treatment conditions.

**Table 2.** Thicknesses of the boride layers obtained on AISI 1018 steel that were treated at 900 °C for a period of 20, 90, 180, or 300 min.

| Time (min) | Temperature (°C) | | |
|:---:|:---:|:---:|:---:|
| | **900** | **950** | **1000** |
| | Layer Thickness (µm) | | |
| 20 | 2.27 ± 0.92 | 9.84 ± 3.86 | 13.29 ± 4.82 |
| 90 | 19.25 ± 5.47 | 20.80 ± 7.00 | 44.88 ± 14.07 |
| 180 | 24.76 ± 8.11 | 32.29 ± 9.95 | 70.06 ± 19.25 |
| 300 | 40.93 ± 11.49 | 48.82 ± 13.99 | 84.76 ± 21.76 |

## 3. Experimental Results

### 3.1. Microstructure

SEM examination of the cross section of the borided samples revealed three zones of interest (Figure 4).

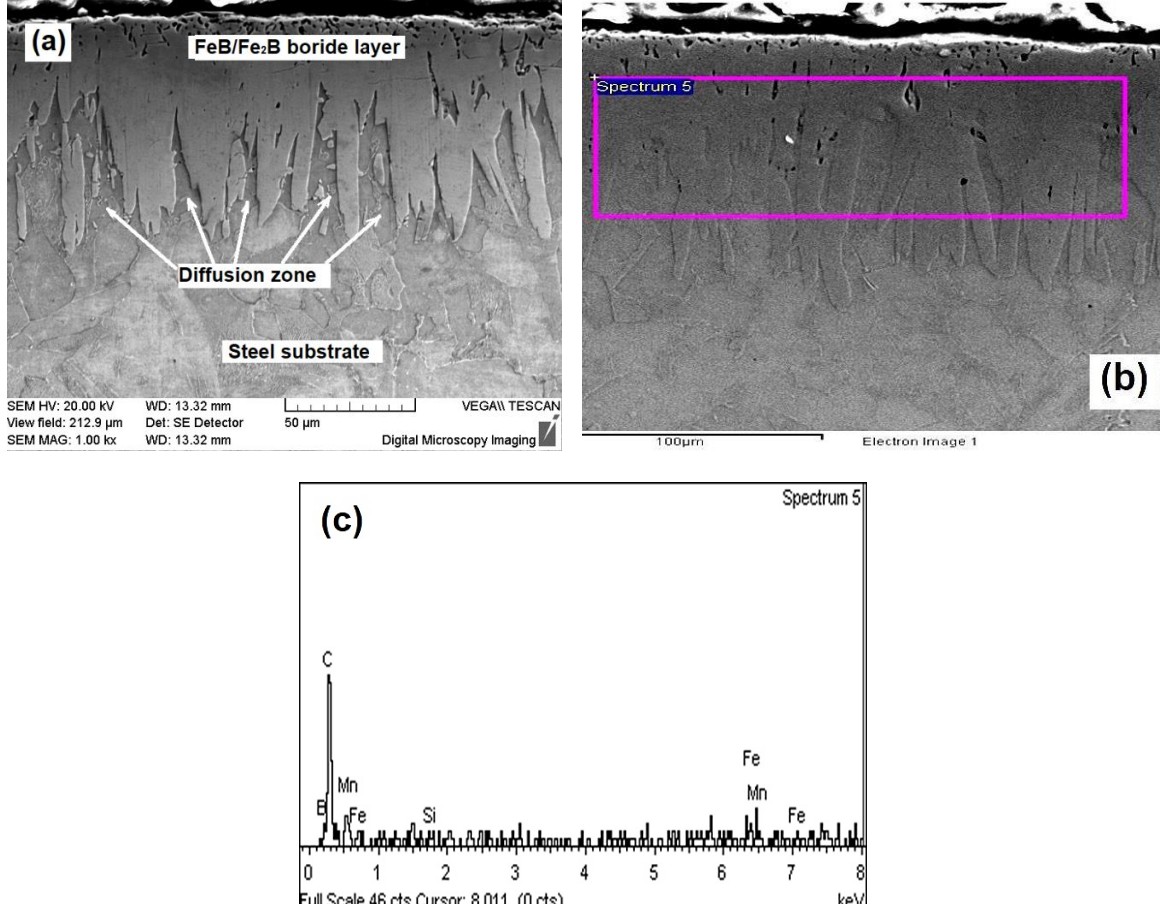

**Figure 4.** Cross section of the samples exposed to 1000 °C during 300 min of treatment where (**a**) shows the three zones of the treated samples, (**b**) indicates the zone evaluated by EDS and (**c**) is the elemental analysis of the boride zone.

The outermost zone was assumed to be a biphasic layer of FeB and Fe$_2$B, and had a sawtooth-like morphology, which is the typical morphology of borided low-carbon steels [35,37,38]. This assumption was corroborated by EDS examination as shown in Figure 4b,c, were it is possible to observe the

presence of boron in the EDS pattern. The second zone was determined to be a diffusion zone, and the substrate (third zone) was found to be unaffected by the diffusion process.

The sawtooth-like microstructure of the boride layer on the low-carbon steel was a product of the diffusion process, which resulted in strongly anisotropic growth in which the preferential growth of the $Fe_2B$ phase occurred in the [001] crystallographic direction [12]. Several studies have established that the orientation of the grains within a boride layer on a polycrystalline substrate seems to be mainly determined by growth kinetics rather than the grain orientation of the substrate material [38–40].

### 3.2. Characterization Results

The layer thickness results that were achieved by following the methodology illustrated in Figure 2 are presented in Table 2 and Figure 5 according to the treatment conditions.

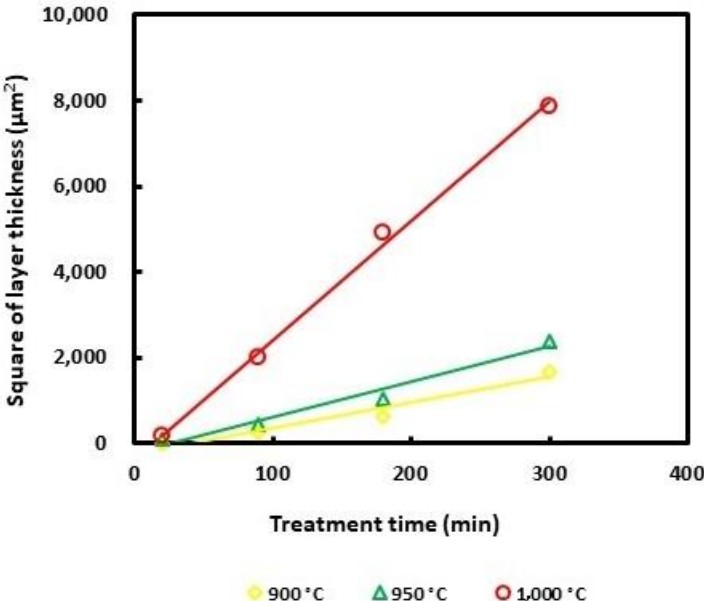

**Figure 5.** Behavior of the layer thickness as a function of the treatment duration.

According to the results depicted in Table 2 and Figure 5, the layer growth can be represented by a parabolic function (Equation (1)), as has been previously explained. The slopes of the lines shown in Figure 5 represent the constant of growth rate $K$, which indicates controlled growth [2,8,12,41]. Additionally, the thickness of the layers increased in response to an increase in the treatment duration and temperature; nevertheless, the temperature was found to have the most substantial influence on layer thickness (see Figure 5). The XRD pattern presented in Figure 6 corresponds to the sample that was subjected to a 950 °C treatment for a period of 5 h.

As is shown in Figure 6, after the treatment, the samples were covered with a primarily biphasic layer comprising FeB and $Fe_2B$, which respectively had orthorhombic and tetragonal crystalline structures, and a crystallographic direction [001] that, as was previously established, has been reported to be the 036-1332 and 00-032-0463, which represent the patterns of $Fe_2B$ and FeB phases, respectively. The strongest reflections were (1,0,1) for FeB, where $2\theta = 45.02°$, (2,1,0) and (2,1,1) for FeB/$Fe_2B$, where $2\theta = 48°$, and small extents (1,1,0) and (1,1,2) for the $Fe_2B$ layer. These results indicate that the layers formed on the surface of the samples following the process of boriding were biphasic layers that were mainly comprised of FeB/$Fe_2B$ phases; this observation is typical of low-carbon steel that has been subjected to boriding.

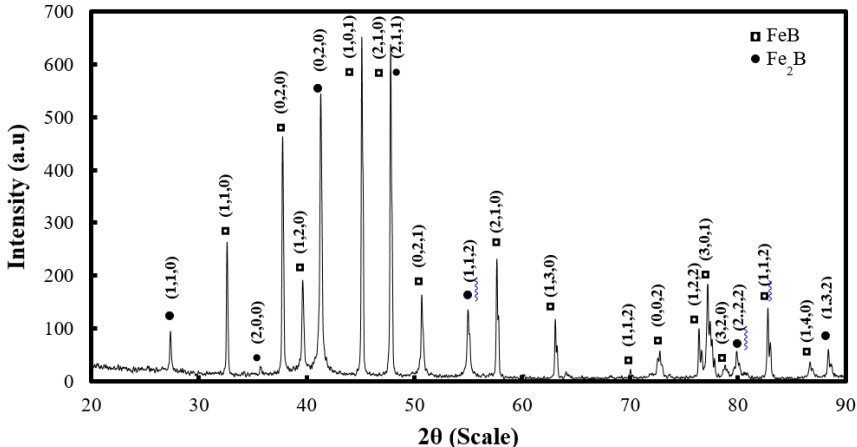

**Figure 6.** XRD pattern of Sample 2, which was subjected to a 950 °C of treatment for a period of 300 min.

According to the microindentation tests, the hardness increased from 1.23 GPa at the level of the AISI 1018 substrate, to 18.89 GPa at the level of the boride layers. These hardness values were compared to those reported in the literature for low-carbon steels subjected to boriding [10,42,43].

## 4. Probability Distribution for Boride Layer Thickness

The best-fit probability distribution for the thickness of a boride layer on carbon steel was investigated by analyzing the data that were obtained from the samples with different treatment times, i.e., 20, 90, 180, and 300 min. The PDFs selected to fit the experimental data were normal distribution, log-normal distribution, gamma distribution, and GEV l istribution. The PDF that best fits the experimental data was determined by using EasyFit Profesional Software V. 5.6 [44]. Table 3 shows how the non-parametric mean and standard deviation of boride layer thickness increased as the treatment duration and temperature increased.

**Table 3.** Non-parametric mean and standard deviation results for the boride layer thickness.

| Time (s) | Temperature (°C) | | | | | |
|---|---|---|---|---|---|---|
| | 900 | | 950 | | 1000 | |
| | Mean (µm) | Standard Deviation (µm) | Mean (µm) | Standard Deviation (µm) | Mean (µm) | Standard Deviation (µm) |
| 1200 | 2.27 | 0.92 | 9.84 | 3.86 | 13.29 | 7.82 |
| 5400 | 19.25 | 5.47 | 20.80 | 7.00 | 44.88 | 14.07 |
| 10,800 | 24.76 | 8.11 | 32.29 | 9.95 | 70.06 | 19.25 |
| 18,000 | 40.93 | 11.49 | 48.92 | 13.99 | 84.76 | 27.93 |

One can see that, on average, these variables influence boride layer formation, as described by Equations (1)–(3), and has been frequently reported in literature [1]. The fact that boride layer growth on carbon steel resulted in a sawtooth-like shape means that layer growth was not homogeneous. Therefore, the standard deviation also increased as the treatment duration increased. The fitted results are presented in Tables 4–6. The parameters obtained for each distribution were determined by using maximum likelihood estimation, whereas the best probability function was selected based on the value obtained via the Kolmogorov-Smirnov test [21,44,45].

**Table 4.** Probability density function fitting results for the observed data for samples exposed to 900 °C of treatment.

| Distributions | 900 °C | | | |
|---|---|---|---|---|
| | **1200 s** | **5400 s** | **10,800 s** | **18,000 s** |
| Normal (μ, σ) * μ = mean and σ = standard devation | *p*-value = 0.097 μ = 2.26 σ = 0.92 | *p*-value = 0.054 μ = 19.25 σ = 5.47 | *p*-value = 0.055 μ = 24.76 σ = 8.11 | *p*-value = 0.035 μ = 40.92 σ = 11.49 |
| Log-normal 3P ** (μ, σ, γ) μ = mean of the variable's natural logarithm, σ = standard deviation of the variable's natural logarithm γ = location parameter | *p*-value = 0.040 μ = 0.96 σ = 0.33 γ = 0.48 | *p*-value = 0.047 μ = 4.59 σ = 0.06 γ = −79.43 | *p*-value = 0.056 μ = 4.90 σ = 0.06 γ = −110.61 | *p*-value = 0.042 μ = 5.70 σ = 0.04 γ = −260.25 |
| Gamma 3P *** (μ, σ, γ) μ = shape parameter, σ = scale parameter γ = location parameter | *p*-value = 0.051 μ = 6.49 σ = 0.36 γ = −0.02 | *p*-value = 0.044 μ = 79.01 σ = 0.61 γ = −29.61 | *p*-value = 0.055 μ = 78.00 σ = 0.92 γ = −47.48 | *p*-value = 0.049 μ = 101.05 σ = 1.17 γ = −77.93 |
| GEV (α, β, ξ) α = location parameter β = scale parameter ξ = shape parameter | *p*-value = 0.034 α = 1.84 β = 0.72 ξ = 0.02 | *p*-value = 0.039 α = 17.18 β = 5.38 ξ = −0.23 | *p*-value = 0.050 α = 21.73 β = 8.05 ξ = −0.28 | *p*-value = 0.022 α = 37.25 β = 11.99 ξ = −0.35 |

* Normal distribution PDF: $f(x) = \dfrac{\exp\left[\frac{-1}{2}\left(\frac{x-\mu}{\sigma}\right)^2\right]}{\sqrt{2\pi\sigma^2}}$; ** Log-normal distribution 3P PDF: $f(x) = \dfrac{\exp\left[\frac{-1}{2}\left(\frac{\ln(x-\gamma)-\mu}{\sigma}\right)^2\right]}{(x-\gamma)\sigma\sqrt{2\pi}}$; *** Gamma distribution 3P PDF: $f(x) = \dfrac{(x-\gamma)^{\mu-1}}{\sigma^\mu\Gamma(\mu)}\exp\left(\frac{-x-\gamma}{\sigma}\right)$.

**Table 5.** Probability density function fitting results for the observed data for samples exposed to 950 °C of treatment.

| Distributions | 950 °C | | | |
|---|---|---|---|---|
| | **1200 s** | **5400 s** | **10,800 s** | **18,000 s** |
| Normal (μ, σ) * | *p*-value = 0.062 μ = 9.83 σ = 3.76 | *p*-value = 0.063 μ = 21.48 σ = 7.13 | *p*-value = 0.056 μ = 32.29 σ = 9.45 | *p*-value = 0.064 μ = 48.92 σ = 13.45 |
| Log-normal 3P ** (μ, σ, γ) | *p*-value = 0.030 μ = 2.80 σ = 0.22 γ = −7.11 | *p*-value = 0.068 μ = 4.79 σ = 0.06 γ = −100.04 | *p*-value = 0.039 μ = 4.37 σ = 0.12 γ = −47.41 | *p*-value = 0.093 μ = 3.84 σ = 0.30 γ = 0.01 |
| Gamma 3P *** (μ, σ, γ) | *p*-value = 0.029 μ = 9.12 σ = 1.25 γ = −1.61 | *p*-value = 0.072 μ = 67.03 σ = 0.87 γ = −37.15 | *p*-value = 0.039 μ = 23.83 σ = 1.94 γ = −14.16 | *p*-value = 0.090 μ = 13.22 σ = 3.69 γ = −0.01 |
| GEV (α, β, ξ) | *p*-value = 0.026 α = 8.23 β = 3.36 ξ = −0.11 | *p*-value = 0.049 α = 18.95 β = 7.20 ξ = −0.18 | *p*-value = 0.033 α = 28.53 β = 9.07 ξ = −0.19 | *p*-value = 0.039 α = 44.75 β = 14.19 ξ = −0.37 |

**Table 6.** Probability density function fitting results for the observed data for samples exposed to 1000 °C of treatment.

| Distributions | 1000 °C | | | |
| --- | --- | --- | --- | --- |
| | 1200 s | 5400 s | 10,800 s | 18,000 s |
| Normal (μ, σ) * | *p*-value = 0.056<br>μ = 13.29<br>σ = 4.82 | *p*-value = 0.076<br>μ = 44.87<br>σ = 14.07 | *p*-value = 0.052<br>μ = 70.06<br>σ = 19.25 | *p*-value = 0.049<br>μ = 84.76<br>σ = 21.76 |
| Log-normal 3P **<br>(μ, σ, γ) | *p*-value = 0.044<br>μ = 3.36<br>σ = 0.17<br>γ = −15.89 | *p*-value = 0.051<br>μ = 5.89<br>σ = 0.04<br>γ = −317.5 | *p*-value = 0.075<br>μ = 5.82<br>σ = 0.06<br>γ = −269.40 | *p*-value = 0.049<br>μ = 5.85<br>σ = 0.06<br>γ = −263.38 |
| Gamma 3P ***<br>(μ, σ, γ) | *p*-value = 0.042<br>μ = 13.18<br>σ = 1.34<br>γ = −4.40 | *p*-value = 0.077<br>μ = 65.35<br>σ = 1.77<br>γ = −71.06 | *p*-value = 0.049<br>μ = 67.64<br>σ = 2.35<br>γ = −89.22 | *p*-value = 0.051<br>μ = 63.74<br>σ = 2.73<br>γ = −89.87 |
| GEV (α, β, ξ) | *p*-value = 0.050<br>α = 12.34<br>β = 4.56<br>ξ = −0.17 | *p*-value = 0.049<br>α = 40.27<br>β = 14.75<br>ξ = −0.24 | *p*-value = 0.039<br>α = 62.83<br>β = 19.11<br>ξ = −0.25 | *p*-value = 0.044<br>α = 76.47<br>β = 21.42<br>ξ = −0.25 |

The mathematical expressions for the normal, log-normal, and gamma distribution PDFs presented above were obtained from reference [44].

It is notable that the PDF that best fit the observed data (i.e., *p*-value ≥ 0.05) was the GEV distribution. The boride layer thickness corresponding to a 900 °C, 20 min treatment is plotted and depicted in Figure 3 as a histogram; the results were also compared to those of the normal, log-normal, gamma, and GEV distributions. Analysis of the fitting results also revealed that the shape of the GEV distribution varied according to the treatment duration, and tended to be skewed when the treatment duration was longer (see Figures 7 and 8). The above-mentioned observations indicate that the histogram of the boride layer thickness data were right-skewed because the boride layer thickness increased with increasing treatment duration. Figure 8 illustrates the time evolution of the PDF, showing that a longer treatment duration corresponded to a larger skew to the right; thus, a longer treatment duration corresponded to a larger standard deviation.

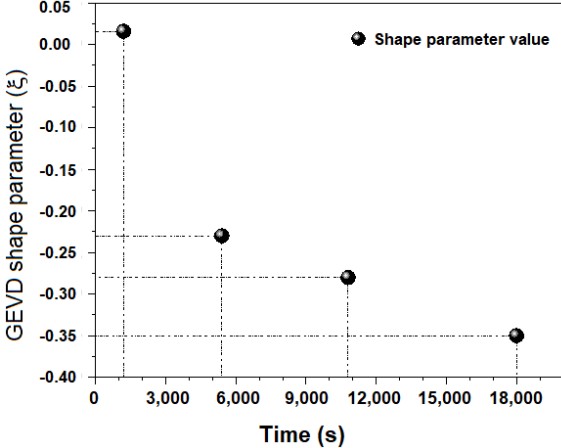

**Figure 7.** GEV distribution shape evolution for a 900 °C treatment.

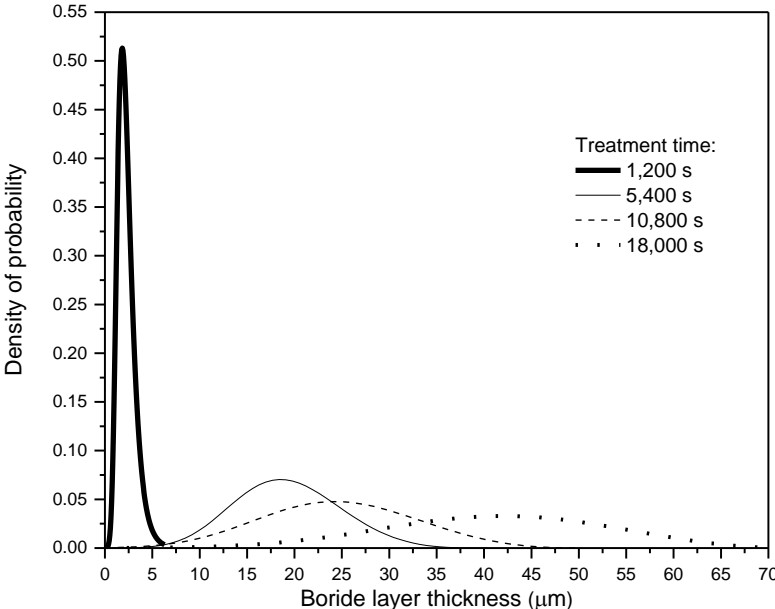

**Figure 8.** Fitted GEV distribution results as a function of treatment time (900 °C treatment).

It is important to remember that the boride layer that is generated on carbon steel has a sawtooth-like morphology because of the diffusion of boron in the [001] direction in the Fe$_2$B phase; this results in a maximum density of boron atoms in this direction, which is the direction of minimum diffusion resistance [38]. The results of implementing a higher temperature, the relationship between boride layer thickness and temperature, and the larger standard deviation results indicate that, at higher temperatures, the diffusion coefficient of boron tends to increase, as has been previously reported [1,38]. Therefore, these findings are consistent with the observations of boride layer formation (i.e., the resulting sawtooth-like morphology), which support the fact that, on average, dispersion increased along with the thickness.

## 5. Stochastic Modeling Approach

### 5.1. Basics of Markov Chain Model

In mathematics, a discrete stochastic process that complies with Markov's property, i.e., if the history of the system is known up to the current moment, the present state summarizes all of the relevant information to describe the probability of the future state [46,47]. A Markov chain is a sequence of random variables, and it is a particular type of discrete stochastic process in which the probability of an event occurring is only dependent on the immediately preceding event. This memoryless feature is typically referred to as a Markov property, which can be expressed as follows:

$$P(A3|A1 \cap A2) = P(A3|A2)$$

This means that, to know the probability that event $A3$ happens, it is not necessary to know the probability of event $A1$; it is only necessary to know the probability of event $A2$.

A Markov chain is determined to be homogeneous when each state that is given in a single step is not dependent on time. Otherwise, if the transition between states is dependent on time, it can be referred to as a non-homogeneous Markov chain. Caleyo et al. [33] proposed a model based on the solution of the forward Kolmogorov equations (see Equation (6)), which were modified to discretize the corrosion pit depth. In this study, it was possible to discretize the boride layer thickness and use the same principle implemented by Caleyo and colleagues [33,34].

In a Markov process defined by the differential equation described by Equation (6), the conditional probability of the transition from the mth state to the nth state ($n \geq m$) for the time interval $(t_0, t)$, i.e., $p_{mn}(t_0, t) = P\{D(t) = n | D(t_0) = m\}$, can be obtained according to the Equation presented on page 304 in reference [46]:

$$p_{mn}(t_0, t) = \binom{n-1}{n-m} e^{-\{\rho(t) - \rho(t_0)\}m} \left(1 - e^{-\{\rho(t) - \rho(t_0)\}}\right)^{n-m} \tag{7}$$

where:

$$\rho(t) = \int_0^t \lambda(t') dt' \tag{8}$$

This means that the boride layer thickness increases over the time interval $t - t_0$ follows a negative binomial distribution $NegBin(r, p)$, with parameters $r = m$, and $p = ps = e^{-\{\rho(t) - \rho(t_0)\}}$.

If the probability distribution of boride layer thickness at time $t_0$ is known, i.e., if $P\{D(t_0) = m\} = p_m(t_0)$ is known, the boride layer thickness distribution at any future moment can be computed by solving the following equation [46,47]:

$$p_n(t) = \sum_{m=1}^{n} p_m(t_0) p_{mn}(t_0, t) \tag{9}$$

It can be observed that the transition probabilities $p_{mn}(t_0, t)$ are dependent on the functions $\lambda(t)$ and $\rho(t)$. Thus, it is necessary to determine a mathematical expression that can accurately represent these functions. For this reason, we propose that the stochastic mean of the boride layer thickness $M(t)$ can be assumed to be equal to the deterministic mean $(t)$ of the boride layer thickness, as follows:

$$(t) = M(t) \tag{10}$$

In references [46,47], it is shown that, if the initial thickness state is $n_i$ at $t = t_i$ such that $D(t_i) = n_i$, then the time-dependent stochastic mean can be represented as:

$$M(t) = n_i e^{\rho(t - t_i)} \tag{11}$$

Additionally, one can assume that a power-law function is an accurate representation of the boride layer thickness deterministic mean that can be expressed as:

$$(t) = \kappa t^\nu \tag{12}$$

where $\kappa$ and $\nu$ are the proportionality and exponent parameters, respectively.

By considering Equations (10)–(12), the value of the function $\rho(t)$ can be represented as:

$$\rho(t) = \ln\left[\kappa t^\nu\right] \tag{13}$$

By taking into consideration Equation (8), it follows that

$$\lambda(t) = \frac{d(\rho(t))}{dt} = \frac{\nu}{t} \tag{14}$$

It is essential to note that the intensity of the Markov process $\lambda(t)$ is inversely proportional to the treatment duration; therefore, the boride layer growth rate decreases over time. It is quite simple to show that the probability parameter $p_s = e^{-\{\rho(t) - \rho(t_0)\}}$ in Equation (7) can be represented as follows:

$$p_s = \left(\frac{t_0}{t}\right)^\nu, \ t \geq t_0 \tag{15}$$

Thus, after summarizing the expressions given in the previous paragraphs, it became possible to obtain the following expression that represents the state transition probability:

$$p_{mn}(t_0, t) = \binom{n-1}{n-m} \left(\frac{t_0}{t}\right)^{vm} \left(1 - \left(\frac{t_0}{t}\right)^v\right)^{n-m} \tag{16}$$

It can be observed that the solution of the Kolmogorov differential system of equations is only dependent on $v$, which is a parameter of the power law. To explain Equation (16), one can use a simple example in which we discretize the boride layer thickness, and one micrometer represents one state. Suppose that a boride layer has 1 µm thickness after 20 min of treatment (1200 s), and that the exponential parameter of the power law $v = 0.5$. With this information, it is possible to determine the probability of transitioning between n and m states. For the sake of illustration, the transition probability for this hypothetical example has been plotted in Figure 9, which shows that the probability of remaining in the same state is close to 0.48, meaning that no transition has occurred; conversely, this means that the probability to transition between two states is approximately 0.13.

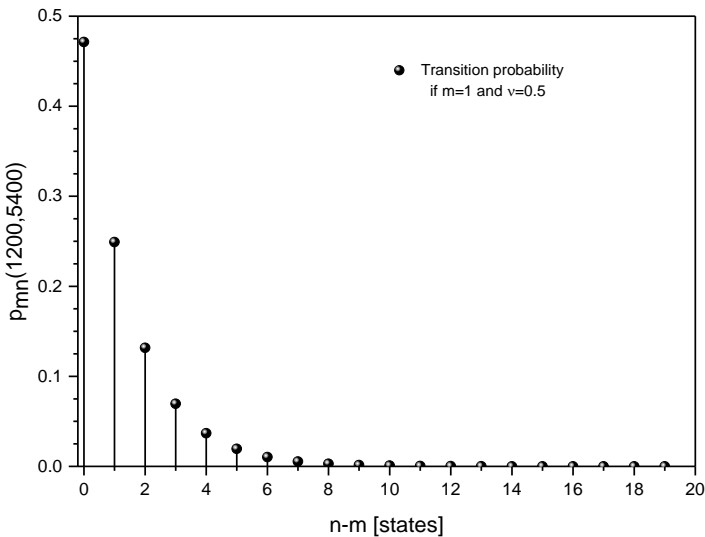

**Figure 9.** Probability of transitioning between States $m$ and $n$ within a time period of 20 to 90 min. (i.e., 1200–5400 s), and with an exponential parameter $v = 0.5$.

### 5.2. Markov Chain Modeling Results

The first step to stochastically estimating the boride layer thickness according to the process described in this paper is to determine a power-law function that represents the phenomena observed under each set of treatment conditions. Table 7 provides values of the above-mentioned parameters, as determined by solving Equation (12), and the coefficient of determination ($R2$) results. From this table, it can be ascertained that a higher treatment temperature corresponded to a lower $v$. This parameter determines the shape of the power-law function, and when the value was relatively low, the resulting curve tended to exhibit asymptotic behavior. Therefore, at higher treatment temperatures, the growth of the boride layer tends to produce a growth rate equivalent to that which occurs with longer exposure times and lower treatment temperatures.

After the power-law parameters were calculated, it was necessary to determine the probability distribution that best represents the probability that the variables in a given state at any start time $t_0$ can be used to determine the probability distribution at any subsequent time $t$. This can be expressed as $p_m(t_0)$ (Equation (9)), and it can be obtained from Table 4. For instance, under the conditions of $t_0 = 1200$ s and a 900 °C treatment temperature, the best-fit distribution is the GEV distribution (see the corresponding p-value results in Table 4). Subsequently, $p_{mn}(t_0, t)$ can be estimated by solving Equation (9). This can be achieved by first solving Equation (16), which only requires knowing the exponential

parameter at the corresponding temperature. Through this process, it is possible to determine the probability distribution that best represents the boride layer thickness at 5400, 10,800, and 18,000 s. Figures 10–12 compare the observed boride layer thickness data and the results that were obtained by using the Markov chain model described in this paper. In these three figures, it can be observed that the mean values of the results obtained via the Markov chain model exhibited a similar trend, and that the observed data and values estimated via the Markov chain model yielded higher boride layer thickness values as time progressed. A similar phenomenon was observed with the standard deviation, as it also increased as time progressed. This indicates that the difference between the relatively thicker boride layers and thinner layers became more significant over time. This is because some peaks of the boride layer began to form later, whereas those that began to form at the beginning continued to grow, implying that the growth rate of the initial peaks may have been higher than that of the newer peaks. Another possible explanation is that, although the newer peaks and initial peaks were subjected to the same treatment conditions, their sizes were different. For this reason, it is possible to accept that the peaks that formed the sawtooth-like boride layer grew at different rates. It should also be noted that the skew of the histogram was reproduced by the Markov chain model. Specifically, with shorter treatment durations, the histograms tended to be skewed to the right.

**Table 7.** Power-law parameters (Equation (12)) for each treatment temperature.

| Temperature (°C) | Exponent Parameter ($\nu$) | Proportionality Parameter ($\kappa$) | Coefficient of Determination ($R2$) |
| --- | --- | --- | --- |
| 900 | 0.89 | 0.25 | 0.96 |
| 950 | 0.67 | 0.06 | 0.95 |
| 1000 | 0.58 | 0.28 | 0.98 |

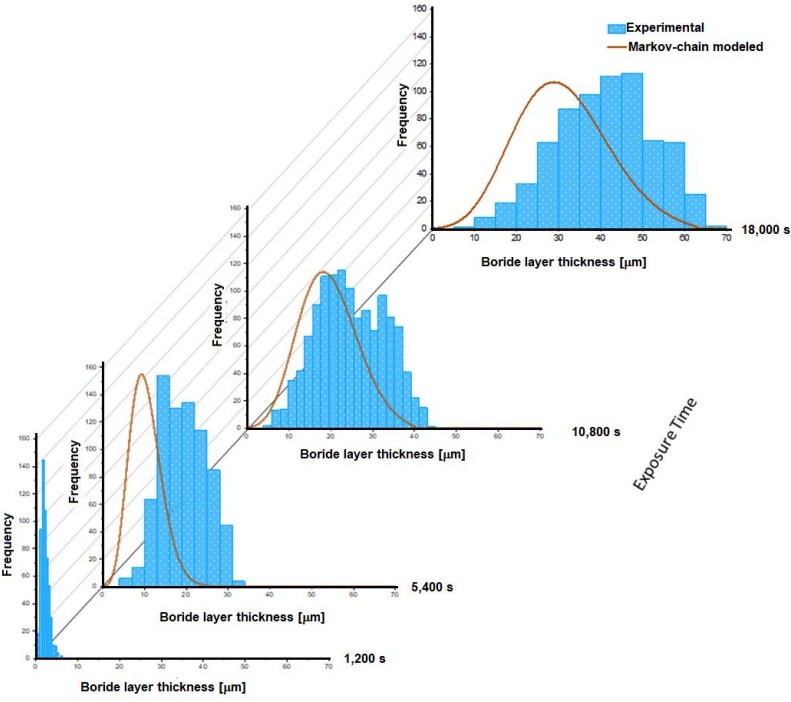

**Figure 10.** Comparison between observed data and results obtained by Markov chain modeling at 900 °C treatment temperature.

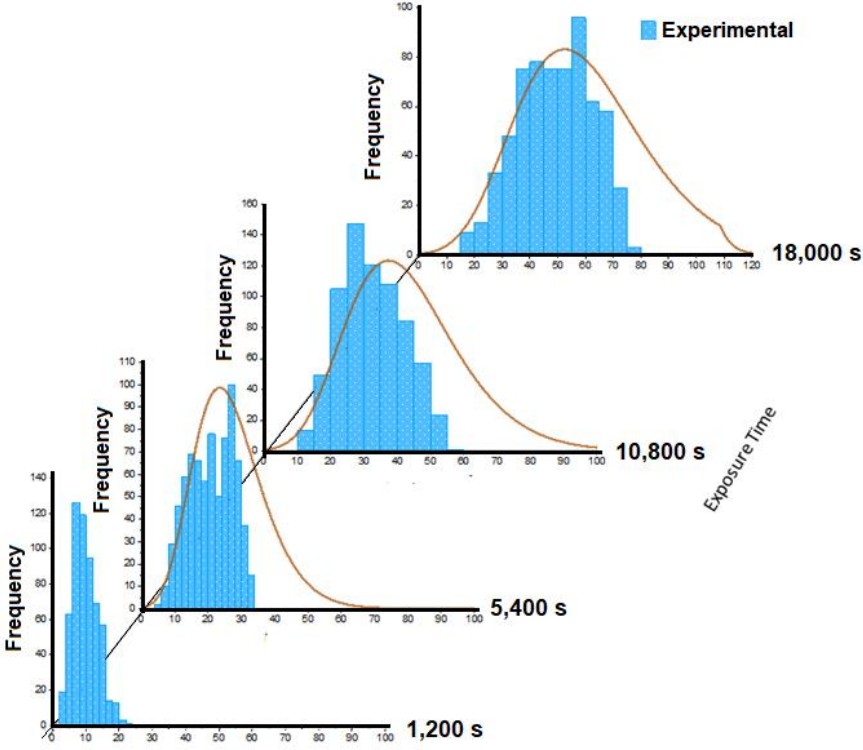

**Figure 11.** Comparison between observed data and results obtained by Markov chain modeling at 950 °C treatment temperature.

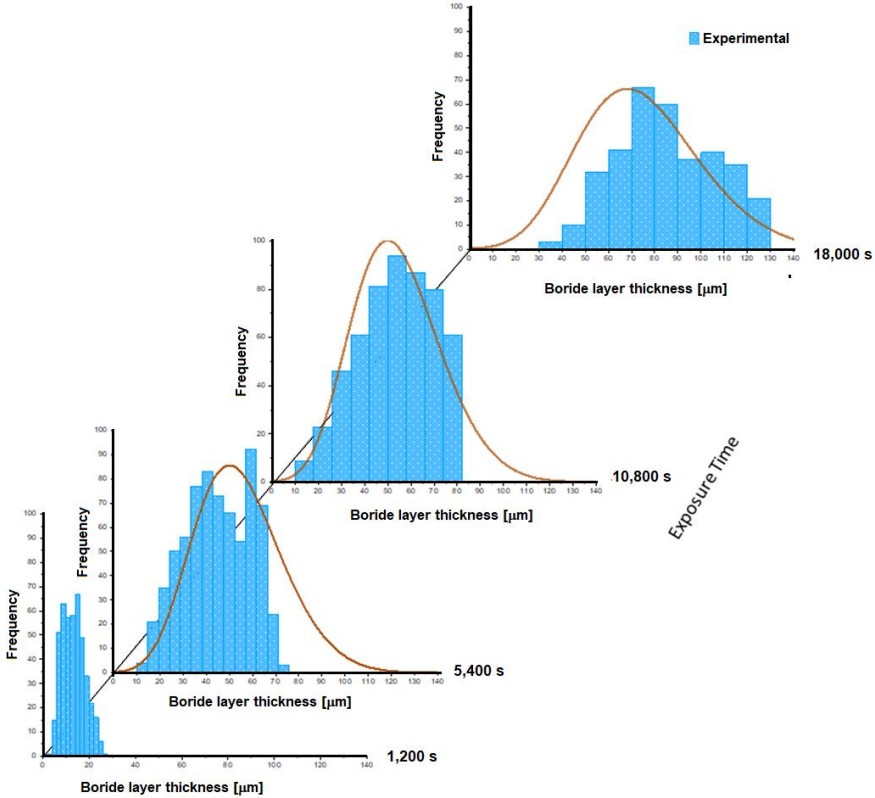

**Figure 12.** Comparison between observed data and results obtained by Markov chain modeling at 1000 °C treatment temperature.

To determine whether the Markov chain model satisfactorily represented the observed data for the boride layer thickness, the mean and variance were compared. The results revealed, with a sufficient level of confidence, that they were reasonably close. In the cases of the mean and standard deviation, the corresponding values were inside of the 95% confidence interval boundaries when the observed data were fitted to a power-law function. These plots are depicted in Figures 13 and 14.

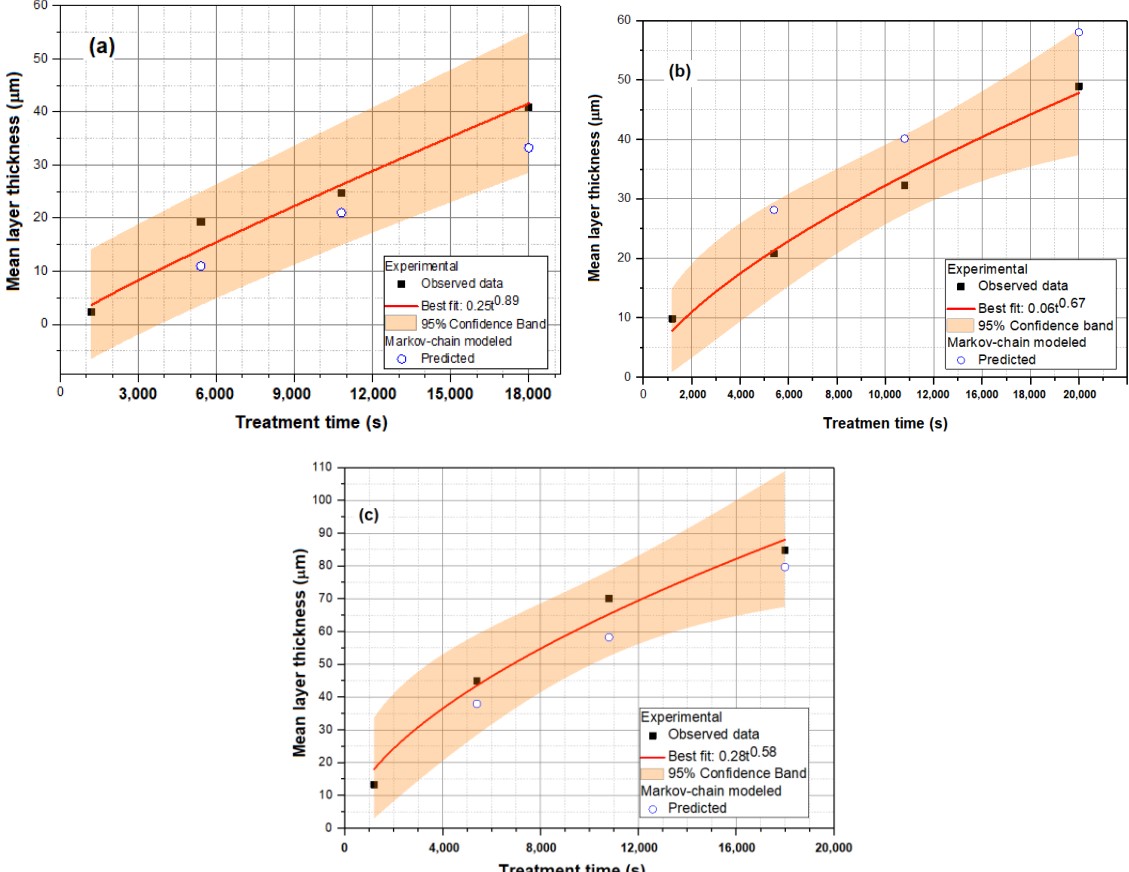

**Figure 13.** Observed and numerically simulated mean boride layer thickness results for treatment temperature of (**a**) 900 °C, (**b**) 950 °C, and (**c**) 1000 °C.

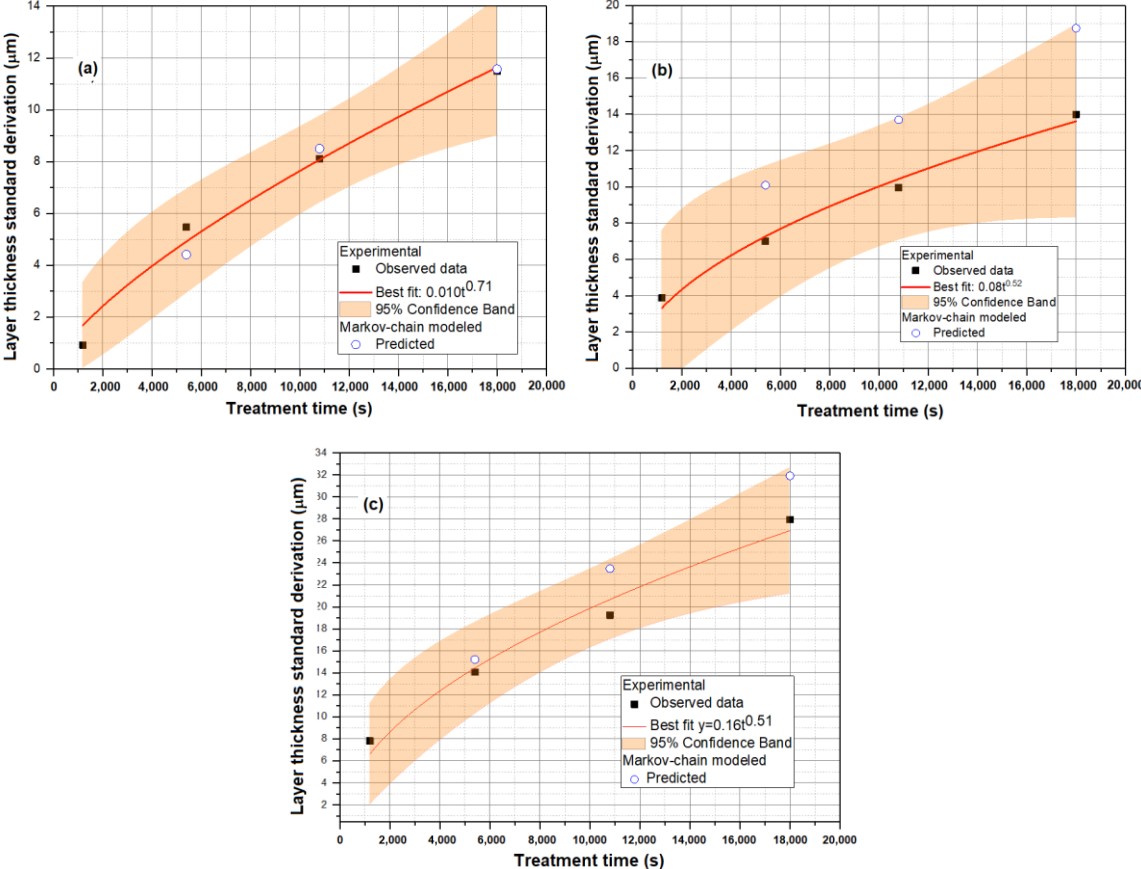

**Figure 14.** Observed and numerically simulated boride layer thickness standard deviation results for treatment temperature of (**a**) 900 °C, (**b**) 950 °C, and (**c**) 1000 °C.

## 6. Conclusions

The conclusions of this study can be summarized as follows:

- A hard surface layer that is mainly comprised of the $Fe_2B$ phase can form when low-carbon steel is subjected to the thermochemical process of boriding;

- The layers that form on low-carbon steels have a sawtooth-like profile, because the boron diffusion process is strongly anisotropic;

- The hardness increased from 1.23 GPa at level of the AISI 1018 substrate to 18.89 GPa at the level of the boride layer, indicating the strong influence of the boriding process on the mechanical properties of the treated materials;

- Since the boriding process yields a non-uniform layer with a sawtooth-like morphology on carbon steel, it is quite easy to confirm that the formation of the boride layer is inherently stochastic. It is only necessary to plot histograms with the layer thickness as the random variable, and the treatment duration as the independent variable;

- The effects of treatment duration on the probability distributions that describe the growth of a boride layer on carbon steel were investigated. The normal distribution, log-normal 3P distribution, gamma 3P distribution, and GEV distribution were selected as theoretical distributions that could potentially represent the heterogeneity of the phenomena. Among these four distributions, the GEV distribution best fit the experimental data;

- The mean and variance of the boride layer formed on the carbon steel tended to increase as the treatment duration increased. This is also evidence of the stochastic nature of the phenomena. This behavior can be explained by the fact that the diffusion coefficient for boron also tended to increase at higher temperatures, and with longer treatment durations. Also this behavior

stimulated the growth and formation of new peaks on the boride layer, demonstrating that, on average, dispersion increased along with the thickness;

- A Markov chain model was developed to simulate boride layer formation as a function of temperature and treatment duration. It was validated by using experimental data. The proposed model is appealing because it obtains the analytical solutions of the system of Kolmogorov's forward equations. Furthermore, the proposed model describes how the thickness of the borided layer changes over time. It was developed by implementing a continuous-time, non-homogeneous linear growth (pure birth) Markov process under the assumption that the Markov chain-derived stochastic mean of the layer thickness equals the deterministic mean of the layer thickness. This assumption allows the transition probability function to be determined if only the exponential parameter is known. Finally, the proposed model allows the boride layer thickness distribution to be predicted at any point in time, which is of paramount importance for automation and optimization processes.

**Author Contributions:** Conceptualization, J.C.V.-A. and E.H.-S.; data curation, I.P.T.-A., G.T.-M., S.I.C.-C. and R.C.-S.; formal analysis, R.C.-E.; investigation, G.T.-M. and S.I.C.-C.; methodology, E.H.-S.; project administration, J.C.V.-A. and E.H.-S.; supervision, E.H.-S.; writing–review and editing, J.C.V.-A. and E.H.-S.

**Funding:** This work was supported by research Grant 20195411 of Instituto Politécnico Nacional in Mexico.

**Acknowledgments:** The authors wish to thank the Center of Nanosciences and Micro-Nano Technologies of the Instituto Politécnico Nacional for their cooperation.

**Conflicts of Interest:** The authors declare no conflict of interest. The funders had no role in the design of the study; in the collection, analyses, or interpretation of data; in the writing of the manuscript, or in the decision to publish the results.

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
