# Peer review of "A Stochastic Model and Investigation into the Probability Distribution of the Thickness of Boride Layers Formed on Low-Carbon Steel"

_coatings, doi:10.3390/coatings9110756_

Round 1

Reviewer 1 Report

Well demonstrated the behavior of non-uniform boride layer
formed on low-carbon steel, regardless of the temperature or duration of treatment and feasibility of its study by the development of a model.

Explain more on the instruments and methods used on characterization of morphology, growth and phase composition of the boride layer. More SEM Images with elemental and phase analysis could have been more supportive.

Author Response

Dear reviewer

The authors appreciate the comments of reviewer 1. First of all, details of instruments, techniques, and methods are given in this new version of the manuscript. Also, it was added a new picture that shows the SEM examination of the cross-section of a sample. Likewise, the elemental composition of the samples was also tested by energy dispersive spectroscopy (EDS), as the reviewer suggests.

All changes in the manuscript are highlighted in yellow to facilitate recognition.

Reviewer 2 Report

Results are well presented and the paper is in agreement with the topic of the journal. In my opinion, the paper could be published in present form.

Author Response

Dear reviewer:

 The authors appreciate the time spent by the reviewer in this process

 Reviewer 3 Report

The article presented here is well written and explores a difficult aspect of boriding which is understanding of thickness. This work provides a useful path for the readers of this journal to explore stochastic models to help explore newer experimental methods to achieve higher uniformity of thickness on low carbon steels.

I just have a couple of questions and a few suggestions for the authors

1. The EDS spectrum does not really provide any useful indication to the reader about the possible formation of FeB/Fe2B mainly because boron is not really detectable by EDS. I would suggest the authors to move it to the supplementary information. 

2. Adding to the EDS its unusual to not observe any peaks from oxygen. Does this mean the authors treated the samples with Ar or any inert gas before performing the  the EDS.

3. I was wondering if the authors might be studying if the hardness might be affected by the change in the ratio of FeB/Fe2B. I would assume that might actually be a change though I understand that this might be hard to control but maybe in the future the authors could look at this study.

4. If possible I would like to know what is the composition of FeB/Fe2B from the powder pattern by performing Rietveld refinements.   

Author Response

Dear Reviewer,

We are very grateful for the time you spent on the revision of our manuscript.

The answers to your comments are described below:

Additionally, please see the attachment, where you will find the manuscript with the respective modifications.

1. The EDS spectrum does not really provide any useful indication to the reader about the possible formation of FeB/Fe2B mainly because boron is not really detectable by EDS. I would suggest the authors to move it to the supplementary information.

If the reviewer considers the EDS spectrum is not necessary to clarify the information, we can consider to remove it from the manuscript. However, this information was suggested by reviewer 1 to provide elemental details of the boride layers. On the other hand, as the reviewer 3 comment, it is quite complicated to detect boron by EDS due to its low molecular weight, nevertheless, as can be observed in the EDS spectrum from Figure 4c, boron was detected just where it has to appear, so, we consider that EDS could supply complementary information about the presence of boron on the surface of treated materials and the authors agree in keeping this figure in the manuscript.

2. Adding to the EDS it’s unusual to not observe any peaks from oxygen. Does this mean the authors treated the samples with Ar or any inert gas before performing the EDS?

The observation of the reviewer is correct due to the oxygen do not appear in the EDS spectrum even when the samples were treated in the absence of inert gas as was mentioned in the manuscript (page 6, line 209), however, as the EDS analysis was realized at the cross-section of the sample (Fig. 4b), so, the oxygen do not interact with this section of the material. Probably if the EDS analysis were made at the surface of the sample, we could observe the presence of that element. On the other hand, as was mentioned in the manuscript (page 6, line 206) samples were covered with 15 mm of powder on each side to prevent oxidation, as have been established by different studies (Ref 36).

3. I was wondering if the authors might be studying if the hardness might be affected by the change in the ratio of FeB/Fe2B. I would assume that might actually be a change though I understand that this might be hard to control but maybe in the future the authors could look at this study.

The comment of the reviewer is correct; the hardness of the boride layer change as a function of the boron concentration. It could be a good challenge at future research to establish the hardness profile as a function of the distance from the surface. Nevertheless, the main objective of this research was to establish the presence of the boride layers using a hardness average of it. Additionally, the manuscript initially said The hardness of each boride phase was evaluated by performing Vickers microindentation with the aid of a CMS-CHV1M Vickers microdurometer” (page 7, lines 227-228) and, this sentence was changed by “The hardness of the boride layers was evaluated by performing Vickers microindentation with the aid of a CMS-CHV1M Vickers microdurometer. This change was implemented to avoid confusion in the interpretation of the zone of the boride layers where the hardness was evaluated in this study.

4.  If possible I would like to know what is the composition of FeB/Fe2B from the powder pattern by performing Rietveld refinements.

As the reviewer suggests, the Rietveld refinement method could provide more information about the composition of the FeB/Fe2B phases. Unfortunately, it is not possible for us, to realize the analysis by now, probably we consider to accomplish this analysis for future studies. We hope it could be enough with the information provided from the XRD analysis and the comparison to the results shown in the literature, as shown in references 10, 41, and 42.